# The Role of Gut Bacteriome in Asthma, Chronic Obstructive Pulmonary Disease and Obstructive Sleep Apnoea

**DOI:** 10.3390/microorganisms10122457

**Published:** 2022-12-13

**Authors:** Andras Bikov, Silvano Dragonieri, Balazs Csoma, Carmen Mazzuca, Panaiotis Finamore, Giulia Rocchi, Lorenza Putignani, Michele Guarino, Simone Scarlata

**Affiliations:** 1Wythenshawe Hospital, Manchester University NHS Foundation Trust, Southmoor Road, Manchester M23 9LT, UK; 2Department of Respiratory Diseases, University of Bari Aldo Moro, 11 Piazza G. Cesare, 70124 Bari, Italy; 3Department of Pulmonology, Semmelweis University, 25-29 Tömő Str, 1083 Budapest, Hungary; 4Unit of Internal Medicine, Campus Bio-Medico University Hospital, 00128 Rome, Italy; 5Division of Allergy, University Department of Pediatrics, Bambino Gesù Children’s Hospital, 00165 Rome, Italy; 6Department of Science and Engineering for Human and the Environment, Campus Bio-Medico University of Rome, 00128 Rome, Italy; 7Department of Diagnostics and Laboratory Medicine, Bambino Gesù Children’s Hospital, 00165 Rome, Italy; 8Unit of Gastroenterology, Campus Bio-Medico University Hospital, 00128 Rome, Italy

**Keywords:** microbiome, gut microbiome, bronchial asthma, chronic obstructive pulmonary disease, obstructive sleep apnea, inflammation, hypoxemia

## Abstract

The human body contains a very complex and dynamic ecosystem of bacteria. The bacteriome interacts with the host bi-directionally, and changes in either factor impact the entire system. It has long been known that chronic airway diseases are associated with disturbances in the lung bacteriome. However, less is known about the role of gut bacteriome in the most common respiratory diseases. Here, we aim to summarise the evidence concerning the role of the intestinal bacteriome in the pathogenesis and disease course of bronchial asthma, chronic obstructive pulmonary disease, and obstructive sleep apnea. Furthermore, we discuss the consequences of an altered gut bacteriome on the most common comorbidities of these lung diseases. Lastly, we also reflect on the therapeutic potential of influencing the gut microbiome to improve disease outcomes.

## 1. Introduction

Despite advances in their management, chronic disorders of the respiratory system, such as asthma, chronic obstructive pulmonary disease (COPD) and obstructive sleep apnoea (OSA) are still common causes of overall morbidity and mortality worldwide. It is becoming widely acknowledged that the development and the course of these disorders are the results of a complex interaction of genetic, epigenetic, and environmental factors. The same factors may also affect the composition of the gut microbiome which in turn may affect the course of these respiratory diseases and associated disorders, such as cardiovascular disease [1]. Understanding the role of gut microbiome in chronic respiratory diseases could be clinically important as gut dysbiosis may serve as a treatable trait.

The microbiome is a complex ecosystem which dynamically interacts with the host cells, immune and metabolic processes. Although microbiome also consists of viruses, fungi, phages, archaea, protists, and helminths [2], most studies focused on the bacteria, especially in the relation of gut-lung axis [1,3]. However, one must note the significant interrelation between different kingdoms, such as the bacteria and fungi; therefore, they cannot be treated in isolation [1]. Nevertheless, as human studies on patients with chronic respiratory disorders overwhelmingly investigated gut bacteria, the current review will focus on this kingdom. The review will focus on studies in humans with some experimental research on animals is also discussed. However, the review did not include all animal studies on this topic which is the main limitation of this paper.

## 2. Lung-Gut Axis

The effect of gut microbiome on the function of the respiratory system has recently been extensively reviewed [1,3]. However, it is less described that factors contributing to the development of lung diseases (i.e., diet and smoking), or their consequences (i.e., hypoxaemia or sleep fragmentation) may also alter gut microbiome.

Diet strongly influences the composition of gut bacteriome [4]. It is known that due to fragmented sleep and leptin resistance, patients with OSA tend to consume high calorie, carbohydrate- and lipid-reach diet [5]. On the other hand, around 25% of patients with COPD are cachectic [6]. Furthermore, the composition of the rectal and respiratory flora shows a close correlation from the birth [7]. Following up infants with cystic fibrosis, Madan et al., demonstrated that the close link is mainly driven by nutrition [8]. This suggests that dietary interventions may influence the respiratory microbiome and local immunity. Sleep restriction, a hallmark of OSA, may also disrupt gut microbiome most likely via increasing the appetite [9].

Cigarette smoking is the main risk factor for development of COPD and hampers asthma control. It can alter gut microbiome as well. Active smoking is associated with increased abundances of *Bacteroidetes*, and decreased abundances of *Firmicutes* [10]. In line with this, smoking cessation increases the abundance of *Firmicutes* and *Actinobacteria* and decreases the abundance of *Bacteroidetes* [11]. However, regarding *Proteobacteria* the results were contradictory. On one hand Lee et al., reported their lower abundance in active smokers [10]. On the other hand, smoking cessation led to the reduction of *Proteobacteria* in stool samples [11]. As the latter study did not include never-smokers, the discrepancies between the two investigations cannot be fully explained.

Air pollution, most particularly fine particulate matter (PM_2.5_), oxides of nitrogen and sulfur, ozone and heavy metals contribute to the development and worsening of COPD [12]. Environmental chemicals may affect gut bacteria through various mechanisms, including modifying their metabolism directly or indirectly following conjugation in liver, causing dysbiosis, and interacting with bacterial products [13]. Most recently, Li et al., have reported that PM2.5 was related to alterations in gut microbiome in healthy subjects [14]. Nitric oxide exposure led to a decrease in the abundance of *Clostridium leptum* group and *Faecalibacterium prausnitzii* and an increase in the abundance of the *Dialister genus*, *Escherichia coli*, *Enterococcus faecalis*, and *Proteus mirabilis* in human fecal samples [15]. Ozone exposure was related to lower bacterial diversity [16]. Finally, the effect of heavy metals on gut microbiome has been extensively reviewed by Claus et al. [13]. 

Obstructive sleep apnoea and in advanced stage, COPD is characterized by chronic hypoxaemia. Some of the gut bacteria, such as *Actinobacteria* are oxygen-sensitive [17], and hypoxaemia may lead to reduced growth of such organisms. A significant number of patients with severe to very severe COPD have chronic hypercapnia [18], and some patients with OSA display blunted ventilatory response resulting in hypercapnia. There is some evidence that hypercapnia is associated with gut dysbiosis in a bidirectional way. On one hand, experimental alterations of gut bacteriome with antibiotics or fecal transplantation blunted the respiratory response to hypercapnia in rats [19]. On the other hand, combined hypoxaemia and hypercapnia has led to alternations in gut microbiome in mice [20,21].

Medications may also affect gut microbiome [22]. Most particularly for respiratory disorders, glucocorticosteroids [23] and antibiotics [24] are known to cause dysbiosis. As patients with respiratory diseases, especially COPD and OSA, tend to suffer from multiple comorbidities, the effect of medications needs to be considered when interpreting the results. Unfortunately, most case–control studies have not adjusted for this effect which could be a potential reason for discrepancies.

Gut microbiome directly influences the local, systemic and distal organ immune responses. On one hand, gut bacteria directly interact with immune cells which process is important for immune cell maturation and polarisation [25]. On the other hand, bacterial metabolites, such as lipopolysaccharides (LPS), short-chain fatty acids (SCFA) may enter the systemic circulation and lymphatic system affecting airway inflammation [26,27]. This relationship may either be deleterious or protective. On one hand, segmented filamentous bacteria induce polarisation of lung T helper (Th) cells towards the pro-inflammatory Th17 type [27,28]. On the other hand, germ-free (GF) animals have worse outcomes against respiratory tract infections, suggesting a protective role of gut bacteria [29,30,31]. The process how gut bacteria may alter lung immunity has been extensively reviewed by Bingula et al. [3].

## 3. Asthma and Gut Microbiome

Asthma affects more than 300 million people worldwide. It is characterised by variable symptoms and airflow limitation that are driven by airway hyperreactivity and chronic airway inflammation [32]. However, the airway inflammation is heterogenous, and different inflammatory processes may lead to the same symptoms [33]. Whilst the type 2 inflammatory pathways are well described and can be addressed by targeted therapies, non-type 2 asthma remains a clinical challenge. 

Gut bacteriome may play a role in the development of atopic asthma [34]. According to the hygiene hypothesis, decrease in the infections in the Western world was followed by the rise in allergic and autoimmune diseases [35]. Indeed, GF mice exhibit a type 2 inflammatory phenotype [36] and have less CD4 T cells in the lamina propria with a higher Th2:Th1 ratio [37]. However, this is complicated by the fact that GF mice have lower CXC chemokine receptor 2 (CXCR2) expression and hence reduced mast cell migration towards the intestinal mucosa suggesting more complex mechanisms [38]. Nevertheless, early exposure to intestinal microbiota reduced the levels of invariant natural killer (iNKT) cells which produce interleukin (IL) 4 and IL-13 with a consequent induction of isotype switching to immunoglobulin E (IgE) in GF animals [39]. In addition, gut microbiota acts on the development and function of regulatory T (Treg) cells by expressing an outer membrane pili-like protein. This protein induces Tregs which suppress the excessive activation of Th2 cells involved in asthma [40].

*Bacteroidetes* are known to ferment dietary fibres into SCFA which seem to be key mediators in the gut-lung axis (Figure 1). SCFA increase IgA production through stimulating dendritic cells in the Peyer’s patches to activate B cells to switch classes [41]. In their elegant experiment, Trompette et al., treated mice with high- and low-fibre diet. They reported an increase in circulating SCFA levels following high-fibre diet which resulted in dampening of the type 2 inflammation [26]. Importantly, the challenge also resulted in decrease in airway hyperresponsiveness [26].

In the Canadian Healthy Infant Longitudinal Development Study, it has been demonstrated that infants who develop atopy and wheeze had lower levels of *Faecalibacterium*, *Lachnospira*, *Veillonella* and *Rothia* genera during the first 100 days of life. In addition, lower levels of SCFA were measured in these children [42]. As a follow up of this observation, the authors inoculated GF mice with these bacteria and reported reduction in airway inflammation [42]. Investigating 690 children, Stokholm et al., reported that at 1 year age, the relative abundance of *Faecalibacterium*, *Bifidobacterium*, *Roseburia*, *Alistipes*, *Lachnospiraceae incertae sedis*, *Ruminococcus* and *Dialister* was inversely related, whilst the abundance of *Veillonella* was directly correlated with the development of asthma at age 5 years but only in children of asthmatic mothers. This suggests that specific alterations of gut bacteriome during the first year of life can trigger the inherited asthma risk [43]. Investigating gut microbiome in 2–12 months children in the Protection against Allergy: Study in Rural Environments cohort, bacteria producing butyrate were in an inverse relationship with asthma risk at school age. The study highlighted the protective effect of farm exposure for asthma development as well [44].

A few case–control studies investigated gut microbiome in asthma. Chiu et al., evaluated 35 children with asthma and 26 healthy controls. They found lower abundance of *Firmicutes* in asthma, but there was no difference in bacterial diversity [45]. Demirci et al., investigated 92 children with asthma and 88 healthy controls. In their targeted analysis, they reported lower abundances of *Akkermansia muciniphila* and *Faecalibacterium prausnitzii* [40]. These bacteria previously shown to be associated with regulatory T cell polarization [46] and therefore the findings of Demirci are suggestive for enhanced inflammatory response. Focusing on 24 asthmatic and 8 non-asthmatic adults, Begley reported decreased *Bacteroidetes*/*Firmicutes* ratio, most particularly reduced abundance of *Bacteroides*, *Enterobacteriaeceae* and higher abundance of *Bifidobacterium* and *Lachnospiraceae* in asthma. These results related to lung function values [47]. In 80 adult volunteers with and 40 without asthma, Okba described higher *Lactobacilli* and *E. coli* abundances; however these results were not related to disease outcomes [48]. In a metagenome-wide association study, Wang et al., compared 36 patients with asthma and 185 control subjects. The authors reported that *Faecalibacterium prausnitzii*, *Sutterella wadsworthensis* and *Bacteroides stercoris* were depleted, whereas *Clostridiums* with *Eggerthella lenta* were over-represented in individuals with asthma. In overall, the authors concluded an increase in SCFA-producing bacteria in asthma [49]. Investigating 47 patients with asthma and 20 healthy subjects, Zou et al., reported that patients with asthma have higher abundance of *Ruminococcus gnavus*, *Bacteroides plebeius*, and *Clostridium clostridioforme* and lower abundance of *Roseburia inulinivorans* and *Clostridium disporicum*. The authors also reported that the gut bacteriome was different in allergic and non-allergic subjects [50]. Comparing 15 patients with severe asthma to 14 with non-severe asthma and 15 healthy controls, Wang et al., demonstrated lower abundance of *Acidaminococcaceae* and higher abundance of *Veillonellaceae* and *Prevotellaceae* in severe asthma. The abundance of *Veillonellaceae* related to lung function [51]. In a very recent clinical trial, Van Engelen et al., modulated gut microbiome by administering antibiotics in patients with asthma. Changes in gut microbiome did not result in any alterations in markers of airway inflammation [24].

In overall, although gut dysbiosis may affect the development of asthma, the results of individual case–control studies were not replicated by other studies. This could be due to significant biological and clinical heterogeneity of asthma. On the other hand, standardization of microbiome analyses is warranted to understand the role of gut bacteriome in asthma.

## 4. COPD and Gut Microbiome

Chronic obstructive pulmonary disease is a chronic, usually progressive disorder of the airways, lung parenchyma and vasculature caused by exposure to fumes and noxious particles, most commonly cigarette smoke [12]. The gut microbiome may alter in patients with COPD due to smoking, environmental exposure, and diet (see Section 2). In addition, patients with severe COPD are particularly susceptible to bacterial infections [52]. An animal study demonstrated that respiratory infections may alter the gut bacteriome, namely, the instillation of intrapulmonary lipopolysaccharide significantly increases the total bacterial count in mice cecum [53].

Unfortunately, the number of studies investigating the gut bacteriome in COPD is low. Comparing 28 patients with COPD to 29 healthy controls, no difference was noticed in the gut bacteriome diversity. However, patients with COPD had an increased abundance of *Streptococcus*, *Rothia*, *Romboutsia* and *Intestinibacter* and decreased abundance of *Bacteroides*, *Roseburia* and *Lachnospira*. The abundance of the members of *Streptococci* and *Lachnospiraceae* inversely related to lung function [54]. Analysing 60 patients with COPD, Chiu et al., reported increased abundances of *Fusobacterium* and *Aerococcus* in more advanced disease. The abundance of *Bacteroides* was related to blood eosinophilia and lung function [55]. In their follow-up study, Chiu et al., reported that patients with rapid lung function decline have a higher abundance of *Firmicutes* and a lower abundance of *Bacteroidetes* and *Alloprevotella*. Along with worsening lung function, the authors reported an increasing abundance of *Acinetobacter* and *Stenotrophomonas* [56].

The gut microbiome may play a role in the occurrence of adverse outcomes during acute exacerbations of COPD. Sprooten et al., have reported an increase in small intestinal permeability during flare-ups suggesting that bacterial products may more likely enter the circulation [57]. In line with this, in patients with COPD exacerbations, increased levels of trimethylamine N-oxide (TMAO), a molecule which is metabolized by the liver from the bacterial product trimethylamine (TMA), were associated with increased long-term mortality [58]. Following up 15 patients with exacerbations during their recovery, Sun et al., have reported significant alterations in individual microbes without any change in their diversity [59].

Due to the low number and cross-sectional nature of the studies in COPD, it is not clear how gut dysbiosis affects the course of COPD. As mentioned in Section 3, dysbiosis may alter airway inflammation; however, this needs to be proven in patients with COPD. On the other hand, gut bacteriome could lead to accelerated lung ageing [60]. Most importantly, changes in the gut microbiome contribute to the development and worsening of comorbidities (see Section 6) which frequently accompany COPD and contribute to its burden [61].

## 5. OSA and Gut Microbiome

Obstructive sleep apnoea (OSA) is common disease which is characterised by the repetitive collapse of the upper airways during sleep. These respiratory events lead to chronic intermittent hypoxaemia and sleep fragmentation and ultimately OSA is a risk factor for cardiovascular, metabolic and cognitive disease [62]. Experimental animal studies show that both hypoxaemia and sleep fragmentation may lead to alterations in gut microbiome [63]. In addition, the effect of Western diet needs to be considered (please see Section 3).

Only a limited number of studies have investigated gut microbiome in OSA so far. In a cohort of two-year old children, Collado et al., have reported that the ratio of *Firmicutes*/*Bacteroides* was higher, whilst the ratio of *Actinobacteria*/*Proteobacteria* was lower in snorers. Snorers also showed lower microbial diversity compared to non-snorers [64]. In line with this, in a limited cohort of children of 2–12 years, Valentini et al., concluded a lower microbial diversity and higher abundance of *Proteobacteria* in patients with OSA [65]. Ko et al., analysed stool samples of 93 adult patients with OSA and 20 controls. They found significant differences only at genus level with decreased relative abundances of a few SCFA-producing bacteria which could lead to epithelial barrier disruption [66]. A further analysis of the same cohort revealed a significant relationship between the *Prevotella* enterotype and OSA [67]. The same group compared 60 patients with OSA and 12 control subjects. In this cross-sectional study, the abundance of *Megamonas*, *Gemmiger*, *Dialister*, and *Oscillibacter* genera were lower in OSA and SCFA-producing bacteria were in an inverse relationship with the presence of hypertension [68]. Investigating 19 adult patients with OSA and 20 controls, Bikov et al., reported a lower abundance of *Actinobacteria* phylum. Although, the abundance of *Proteobacteria*, *Gammaproteobacteria*, *Lactobacillae*, and *Lactobacillus* were related to disease severity, dyslipidaemia and cardiovascular disease, following adjustment on cardiometabolic risk factors, these associations became insignificant. This highlights the importance of adjusting for external factors on gut bacteria [69]. Investigating 32 patients with OSA and 14 healthy controls, Wang et al., concluded that OSA was associated with increased ratio of *Firmicutes*/*Bacteroides*. At genus level, the authors reported that the abundance of *Rikenellaceae* and *Alistipes* increased and *Clostridium_XlVa* decreased in OSA [70]. In patients with type 2 diabetes mellitus (2TDM), the presence of OSA was not associated with altered diversity, and there were only subtle differences, such as increased abundance of *Oscillibacter* and decreased abundance of *Phascolarctobacterium* in patients with OSA [71]. Finally, in patients with OSA, carotid atherosclerosis was associated with lower abundance of *Peptostreptococcaceae* [72].

## 6. Consequences of Altered Gut Microbiome on Frequent Comorbidities of Lung Diseases

The physiological role of human microbiome has been extensively reviewed previously [73,74]. In the current review we will focus on results identified by studies described in Section 2, Section 3 and Section 4 and how these relate on comorbidities.

One of the functions of the healthy microbiome is to maintain the metabolism of the epithelial cells in the gut. In line with this, dysbiosis may lead to disruption in the epithelial barrier. Through the disrupted intestinal barrier, gut microbiota dependent metabolites, bacterial toxins, and other bacterial products can enter the systemic circulation [75]. These bacterial products and metabolites may induce a proinflammatory state in the host, increasing the risk of cardiovascular diseases (CVD), and potentially worsening the inflammation associated with airway diseases.

One of the most important gut-microbiota dependent metabolites is TMA, a product of bacterial choline/phosphatidylcholine metabolism [76]. TMA produced in the intestines enters the host circulation and is oxidised in the liver via flavin monooxygenases into TMAO [73]. Previous studies have confirmed that high blood TMAO levels are associated with heart diseases [77,78] through enhancing atherosclerosis [76], promoting platelet reactivity [79,80], worsening vascular inflammation and increasing inflammasome activation [81,82], and augmenting oxidative stress [83,84]. These pathophysiological processes assessed in vitro and in vivo explain the results of human clinical studies reporting an association between circulating TMAO levels and adverse outcomes in peripheral and coronary artery disease [85], acute coronary syndrome [86], and heart failure [87].

Phenylacetylglutamine (PAG), a derivate of phenylalanine metabolism, is also a gut-microbiota dependent product which is associated with major adverse cardiac events, e.g., heart attack, ischemic stroke, and death [88]. PAG induces platelet hyperresponsiveness and thus increases thrombosis potential through the activation of G-protein-coupled receptors (GPCR), including α- and β-adrenergic receptors [88]. Moreover, Nemet et al., has also proven in the same study that in mice the β-blocker carvedilol reverses the pro-thrombotic effect of PAG [88].

Additionally, besides enhanced production of microbiota-dependent metabolites, dysbiosis can also modify the production and absorption of other intestinal molecules, including primary and secondary bile acids, and SCFA. Secondary bile acids (SBA) are synthetised by gut bacteria via bile salt hydrolysis or 7α-dehydroxylation of primary bile acids, and the production is mainly regulated by the interaction between the host and the gut microbiota. SBAs absorbed in the circulation impact the host physiology in several ways, including GPCRs, farnesoid-X receptors, liver-X receptors and pregnane-X receptors [75]. As SBAs have hormone-like functions, they have been linked to metabolic disorders, for example type 2 diabetes mellitus [89]. In addition, SCFAs (e.g., acetate, propionate, butyrate) are also a product of bacterial processes with physiologic functions, but the disturbances in the absorbed SCFA milieu may have pathologic consequences, such as hypertension, adiposity [90], or worsening of the airway inflammation [18]. SCFAs are produced from dietary fibres through anaerobic fermentation in the intestines, and they act as an energy source for intestinal epithelial cells and have an important role in the regulation of blood pressure and glucose and lipid metabolism. The receptors that mediate the effects in blood pressure regulation are the olfactory receptor 78 (Olfr78) [91] and the G-protein receptor 41 (Gpr41) [92]. The activation of Olfr78 causes hypertension through the renin-angiotensin pathway, while Gpr41 has the opposite effect by relaxing the vascular smooth muscle cells. The role of these receptors and SCFA generation was proven in multiple animal studies, either via SCFA administration in genetic knock-out mice [91] or by fecal transplantation in germ-free mice from human hypertensive or control normotensive donors [93]. However, it is likely that the overall effect of SCFAs on blood pressure is influenced by genetic variations and the cross-talk with other microbiota-dependent plasma metabolites [75].

Furthermore, Gram-negative bacteria, such as *Proteobacteria* produce LPS which can enter the circulation if the host barrier function is not intact. LPS may then induce vascular, systemic and pulmonary inflammation through the activation of surface Toll-like receptors of immune cells [94]. Several mechanistic models performed in cell cultures, animals, and humans proved that the administration of LPS triggers an inflammatory response in the host [95,96,97], and causes dysfunction in certain cells, such as cardiomyocytes [98]. Additionally, a human observational clinical study also confirmed the relationship between gut-derived systemic LPS concentration and CVD risk, namely, the LPS level was found to be predictive for major adverse cardiovascular events in a cohort of patients with atrial fibrillation [99].

Gut bacteria may produce neurotransmitters or their precursors which could affect mood, cognitive function and sleep quality [100]. For instance, *Actinobacteria*, *Lactobacilli* and *Bifidobacteria* can produce gamma-aminobutyric acid (GABA) which is a sleep promoting neurotransmitter [101]. In line with this, Smith et al., have reported an inverse relationship between Actinobacteria and the number of awakenings in healthy volunteers [102]. Very interestingly, feces transplanted from mice exposed to intermittent hypoxaemia induced sleepiness in recipient mice, suggesting that hypoxaemia-induced sleepiness in OSA may be partially mediated by gut microbiome [103]. It has also been reported, that increasing the gut microbiota diversity and the abundance of certain bacterial phyla, such as *Actinobacteria* and *Firmicutes*, is associated with increased cognitive function [104]. Furthermore, psychological factors, e.g., emotional stress or sleep deprivation can also influence the microbiota profile potentially leading to dysbiosis, intestinal inflammation, and increased intestinal permeability [105,106]. The interplay between mental health and the gut microbiome is thus bi-directional, which proposes further research and possible therapeutic potential [107].

## 7. Therapeutic Aspects of Influencing the Gut Microbiome

A number of different preclinical and clinical studies have explored the use of probiotics, prebiotics, dietary components and faecal microbiota transplantation (FMT) to modulate the gut microbiota as therapeutic strategies to influence the gut-lung axis [3,108]. However, the specific efficacy of these supplements in the treatment of respiratory diseases, including asthma, COPD, lung cancer and respiratory infections, has not been clearly defined.

### 7.1. Oral Supplementations for the Treatment of Respiratory Diseases

According to the WHO nutritional guidelines, probiotics are useful supplements containing live microorganisms that, when taken properly, provide important health benefits for an individual [109].

#### 7.1.1. Oral Supplementations in Asthma

Mice fed with high fibre diet had increased circulating levels of SCFA and were protected against allergic airway disease [26]. Similarly in mice, gut inoculation with *Lactobacillus johnsonii* significantly reduced the Th2 response in the lungs [110]. Supplementation of *Ligilactobacillus salivarius* LS01 and *Bifidobacterium breve* B632 on a paediatric population with asthma resulted in a significant reduction in the number, frequency and severity of asthma flare-ups [111]. A study conducted in mouse models argues that a combined nutritional intervention with *Bifidobacterium lactis* BB-12 and the supplementation of nutrients with antioxidant and anti-inflammatory properties (docosahexanoic acid, vitamin C and E) is a rational option to alleviate air pollution-related lung inflammation [112]. Recently, a 3-month randomized controlled study investigated the efficacy of co-administration of *Bifidobacterium lactis* Probio-M8 with conventional therapy in the management of asthma, observing an improvement in asthma symptoms and reduction in exhaled and alveolar nitric oxide levels [113]. More importantly, the therapeutic synergy with the probiotic increased the resilience of the gut microbiome showing significant increases in the potentially beneficial species of *Bifidobacterium animalis*, *Bifidobacterium longum* and *Prevotella*, and decreases in *Parabacteroides distasonis* and *Clostridiales*, compared to the placebo group [113].

Administration of *Lactobacillus rhamnosus* mitigated airway inflammation and bronchial reactivity in an asthma mice model [114]. In addition, obese mice fed with *Lactobacillus gasseri* exhibited better anti-microbial response [115]. Moreover, it was observed that Probio-M8 reduced the duration of flu-like symptoms compared with more prevalent anti-inflammatory effects against lower respiratory tract infections [116].

#### 7.1.2. The Effect of Oral Supplementations on the Upper Respiratory Tract

A randomised, double-blind, parallel, placebo-controlled study evaluated the effects of *Bifidobacterium longum* BB536 on diarrhoea and/or upper respiratory tract disease in 520 children aged 2–6 years [117]. Although BB536 did not exert significant effects against diarrhoea, this multifunctional probiotic was shown to reduce the duration of common upper respiratory tract infections (URTI) by modulating the gut microbiota. Analysis of the gut microbiota at the genus level revealed a significantly higher abundance of bacterial genera with immunomodulatory and anti-inflammatory properties, such as *Faecalibacterium* in the treated group compared to the placebo group, illustrating the potential protective effects of BB536 against respiratory diseases [117]. Improvements in the duration of nasal symptoms and the frequency of URTIs were obtained following daily administration for 12 weeks of *Lactobacillus plantarum* DR7 [118], a strain isolated from cow’s milk that acts on the activation and phosphorylation of protein kinase AMP (AMPK) [119], exerting a certain immunomodulatory protective role against URTIs and influenza virus infections through activation of macrophages [120]. Reduced plasma levels of interferon-γ (IFN-γ) and tumor necrosis factor-α (TNF-α), and an enhancement of anti-inflammatory cytokines (IL-4, IL-10) were also observed, accompanied by reduced levels of oxidative stress, lower expression of CD4 and CD8 cells, and a higher presence of NK cells in adults who received DR7 compared to placebo [118]. Similar results were reported following oral administration of *Lactobacillus casei* Zhang (9 log CFU per day) against URTIs and gastrointestinal disorders, through potential anti-oxidative and immune-modulatory effects [121].

#### 7.1.3. Oral Supplementations in COPD

There is evidence that the use of probiotics may have an effect on COPD thus confirming its connection with the gut microbiota [122,123]. Verheijden showed that intragastric supplementation with *Lactobacillus rhamnosus* and *Bifidobacterium breve* in mice mitigated airway inflammation and alveolar damage [122]. Furthermore, Mortaz et al., revealed that the same two probiotics had analogous anti-inflammatory effect on cigarette smoke-induced inflammation in human macrophages [123]. The intake of a multi-strain probiotic on faecal microbiota composition and bowel movements in patients with COPD treated with antibiotics showed a modest probiotic effect on bacterial subgroups, i.e., an increase in yeasts and a decrease in *Bifidobacteria*, in the treated group compared to placebo [124].

#### 7.1.4. Oral Supplementations in Sleep Disorders

*Lactobacillus fermentum* strain PS150 improved sleep quality in mice [125] by promoting non-rapid eye movement (nREM) sleep [126]. On the other hand, ergothioneine, a metabolite of *Lactobacillus reuteri*, increased REM sleep duration in rats [127]. A prebiotic diet has prolonged both REM and nREM sleep in another study in rats [128]. A recent review on this topic concluded that these alterations could lead to improvement in mental disorders in patients with sleep disorders [129]. Finally, administration of probiotics improved coronary artery changes induced by intermittent hypoxaemia but could not fully mitigate coronary artery disease [130].

### 7.2. Preliminary Evidence on Fecal Microbiota Transplantation on the Gut-Lung Microbiota Axis

Given that intestinal bacterial flora plays an important role in the physiology of the human body, among the possibilities of actively intervening on the intestinal microbiota, either for health or therapeutic purposes, faecal microbiota transplantation (FMT) is gaining popularity, not least because of its apparent ‘radicality’ [131]. As reported earlier, gut dysbiosis influences lung disease via the gut-lung axis. Preliminary studies on FMT have explored the mechanism of rebuilding gut flora on respiratory diseases. Experimentally, it was observed that the intervention of a diet rich in fibre and FMT reduced the severity of pulmonary emphysema in mice with COPD, while also increasing the abundances of SCFAs-producing gut bacteria, such as *Bacteroidaceae* and *Lachnospiraceae* [132]. In another study on smoking induced COPD mice model, fecal transplantation ameliorated COPD. The authors highlighted the protective role of *Parabacteroides goldsteinii* [133]. Similarly, a research group explored the mechanism of gut flora regulation in the host defence against LPS-induced acute lung injury by constructing a model of gut microflora dysbiosis with antibiotic administration and reconstruction of the gut ecology via FMT. Following FMT intervention, a down-regulation of the TLR4/NF-kB signaling pathway was observed in the lung and a reduction in the levels of inflammation and oxidative stress in animals with acute lung injury by restoring the gut microbiota composition [134]. Another study on the gut-lung axis showed that the process of restoring commensal flora by FMT in C57BL/6 mice deprived of the gut microbiota and intranasally infected with *S. pneumoniae* resulted in normalisation of lung bacterial counts and the levels of TNF-α and IL-10 six hours after infection. The study concluded that the gut microbiome is a protective factor against pneumococcal pneumonia by regulating alveolar macrophage function and inflammatory response [135].

It is, however, a more complex method than it may seem, which in fact selectively transfers some strains and others do not and should be precisely defined in terms of procedure, treatment of the bacterial material, selection of the donor and assessment of the risk of pathogenic bacterial contamination. Further future research will be needed to evaluate the efficacy and safety of FMT in modulating the gut microbiota in patients with respiratory diseases.

## 8. Summary

The interplay between the respiratory system and the gut bacteriome is multifactorial (Figure 2). On one hand, it opens an avenue to study gut bacteriome as a treatable trait, especially when reducing cardiovascular risk. On the other hand, it brings a complexity when interpreting and comparing the results of human studies. The effect of respiratory pharmaceutical treatment on gut bacteriome was rarely investigated, whilst non-pharmaceutical approaches, such as oxygen or positive airway pressure were not studied at all. Although oral supplementation and fecal transplantation are promising in animals, they still need to be tested in large randomized controlled trials in humans.

## Figures and Tables

**Figure 1 microorganisms-10-02457-f001:**
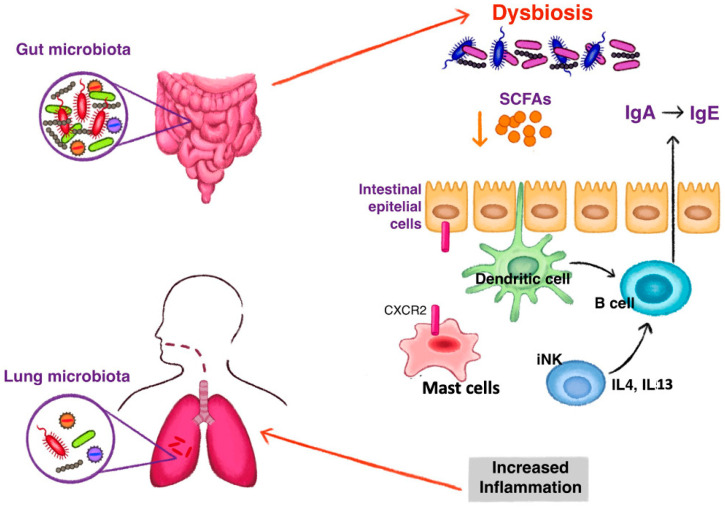
The mechanism of gut dysbiosis leading to the development of type 2 inflammation in asthma. Reduction of short chain fatty acids (SCFAs) induce a class switching of immunoglobulin (Ig) with an increase of fecal IgE acting on dendritic cells. Switching to IgE production is also stimulated by high levels of interleukin (IL) 4 and IL 13, produced by invariant natural killer (iNK) cells under dysbiosis stimuli. Dysbiosis influences the homing of mast cells to the intestine by the expression of CXCR2. Fewer intestinal mast cells and increased blood levels stimulate an inflammatory state observed in asthma.

**Figure 2 microorganisms-10-02457-f002:**
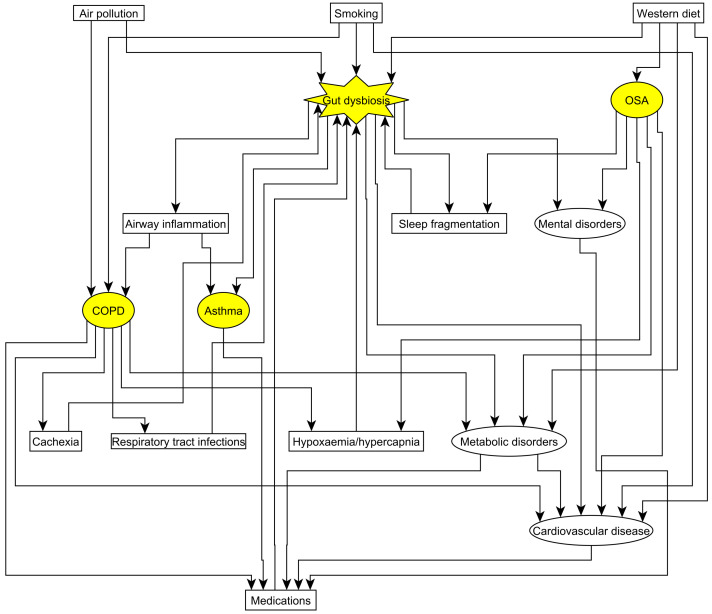
The complex interrelation between asthma, COPD, OSA and gut dysbiosis. For the mechanism in detail, please check the text.

## Data Availability

Not applicable.

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
