# Peer review of "The Role of Gut Bacteriome in Asthma, Chronic Obstructive Pulmonary Disease and Obstructive Sleep Apnoea"

_microorganisms, 2022, doi:10.3390/microorganisms10122457_

Round 1

Reviewer 1 Report

The current manuscript by Bikov et al reviewed the importance of lung-gut axis in three respiratory diseases- asthma, chronic obstructive pulmonary disease, and obstructive sleep apnoea. The manuscript is well-written with the elaborate discussion on the identified sub-topics. But the figures need more editing. Authors are requested to address the following suggestions.

1.     Authors must include a paragraph on the limitations of the review. Similar to other reviews, all information on this specific topic cannot be compiled in a single review. But authors should address some of the critical points left out of the review. This is left at their own discretion.

2.     Authors should consider including one heading on the environmental factors, and how these affect the gut biota, in turn influencing the pulmonary health.

3.     Authors should consider editing the figure 1 to indicate the gut dysbiosis first, on the upper left of the figure and ultimately resulting in the effects on lung microbiota. the "Increased inflammation" and "Dysregulated immune response" in the figure are rather confusing, hence should be edited. There should be some link between dysbiosis and SCFA.

4.     The therapeutic aspect presented in this review should be divided into different types of interventions. An illustration here will be very supportive.

5.    Figure 2, although critical, is very hard to follow. Hence a simplified version will be more appreciated. The different components presented in this figure have not been elaborately discussed in the review. Authors may consider editing the figure to simplify the interrelations.

Author Response

Comment: Authors must include a paragraph on the limitations of the review. Similar to other reviews, all information on this specific topic cannot be compiled in a single review. But authors should address some of the critical points left out of the review. This is left at their own discretion.

Response: Thank you. We believe that the main limitation is that the review focused on human studies with some animal studies discussed. However, we did not include all animal studies. This was discussed at the end of section “1. Introduction”.

Comment: Authors should consider including one heading on the environmental factors, and how these affect the gut biota, in turn influencing the pulmonary health.

Response: Thank you. We have prepared a paragraph on this topic. Please, read 4th paragraph in section “2. Lung-Gut Axis”.

Comment: Authors should consider editing the figure 1 to indicate the gut dysbiosis first, on the upper left of the figure and ultimately resulting in the effects on lung microbiota. the "Increased inflammation" and "Dysregulated immune response" in the figure are rather confusing, hence should be edited. There should be some link between dysbiosis and SCFA.

Response: Thank you very much! The figure and corresponding descriptions have been edited.

Comment: The therapeutic aspect presented in this review should be divided into different types of interventions. An illustration here will be very supportive.

Response: Thank you for your comment. This section has been restructured accordingly.

Comment: Figure 2, although critical, is very hard to follow. Hence a simplified version will be more appreciated. The different components presented in this figure have not been elaborately discussed in the review. Authors may consider editing the figure to simplify the interrelations.

Response: Thank you, we have simplified the figure and explained better the associations in the text.

Reviewer 2 Report

This is an interesting review about the role of the intestinal bacteriome in the most common respiratory diseases.

The authors well summarize the main evidence concerning the role of the intestinal bacteriome in the pathogenesis and disease course of asthma, COPD and OSA.

In addition, they describe the consequences of an altered intestinal bacteriome on the most common comorbidities of these lung diseases and the potential therapeutic role of the gut microbiome.

The review is generally well written and structured. The introduction provides sufficient background and include all relevant references. All paragraphs are well written and structured.

But the quality of the writing and the figure presented could have been much better.

In fact, my only note is on Figure 2. This figure doesn't have a good resolution, it's messed up and it's not well done. Please improve it.

English language and style are quite good. Please do a spell check.

Author Response

Comment: In fact, my only note is on Figure 2. This figure doesn't have a good resolution, it's messed up and it's not well done. Please improve it.

Response: We have modified the figure and improved its resolution.

Comment: English language and style are quite good. Please do a spell check.

Response: We have performed a spell check.

Reviewer 3 Report

Bikov and colleagues have reported pieces of evidence supporting the role of gut microbiota in the most common respiratory diseases. I found the review exhaustive and clear.

I would suggest only modest adjustments.

Major comments:

Lines 69-71: This period is not clear, and should be rephrased to better explain the meaning

Lines 72-79: It is not clear whether the hypercapnia alters the microbiome or vice versa

Line 160-161: In this period a verb seems to be missing or that the period has been somehow truncated

Line 190-191: In which way do Streptococci and Lachnospiraceae contribute to lung function? Detrimental o beneficial way?

Lines 201-203: TMAO is not a product of bacterial metabolism. TMA is produced through bacterial metabolism and is further metabolized to TMAO in the liver.

Minor comments:

Line 53: I would rephrase as follows: “However, it is less described that…”

Line 63: I would put “a hallmark of OSA” between commas

Line 111: remove IL from brackets

Line 117: SCAFs abbreviation can be used as this acronym has been previously explicated in the text

Line 170: I would remove “surprisingly” as it is not a scientific term

Line 190: “Members” should be changed to “members”

Author Response

Comment: Lines 69-71: This period is not clear, and should be rephrased to better explain the meaning

Response: Thank you. We have clarified this in the revised manuscript.

Comment: Lines 72-79: It is not clear whether the hypercapnia alters the microbiome or vice versa

Response: Thank you for the comment. The links seems to be bidirectional. We have modified the text to better clarify this.

Comment: Line 160-161: In this period a verb seems to be missing or that the period has been somehow truncated

Response: Thank you. We have corrected the sentence.

Comment: Line 190-191: In which way do Streptococci and Lachnospiraceae contribute to lung function? Detrimental o beneficial way?

Response: There was an inverse relationship between their abundance and lung function. Due to the cross-sectional nature of the study, the causality is not known. We have clarified this in the manuscript.

Comment: Lines 201-203: TMAO is not a product of bacterial metabolism. TMA is produced through bacterial metabolism and is further metabolized to TMAO in the liver.

Response: Thank you. We have clarified this error in the manuscript.

Comment: Line 53: I would rephrase as follows: “However, it is less described that…”

Response: Thank you. We have corrected it.

Comment: Line 63: I would put “a hallmark of OSA” between commas

Response: Thank you. We have corrected it now.

Comment: Line 111: remove IL from brackets

Response: This is the first time interleukin (IL) abbreviation is introduced. Hence it is in brackets.

Comment: Line 117: SCAFs abbreviation can be used as this acronym has been previously explicated in the text

Response: Thank you. We used abbreviation in the revised version.

Comment: Line 170: I would remove “surprisingly” as it is not a scientific term

Response: Thank you. We deleted “surprisingly”.

Comment: Line 190: “Members” should be changed to “members”

Response: Thank you. This has been corrected.

Round 2

Reviewer 1 Report

Authors tried their best to address the comments.